MHPSS; cognitive coping; veterans; psychosocial intervention; Ukraine

**Corresponding author:**
Amanda Nguyen;
Email: ajnguyen@virginia.edu

# The added benefit of including cognitive coping in brief psychosocial interventions: A randomized controlled trial among veterans and family members in Ukraine

Amanda Nguyen[1] , Tara Russell[2], Stephanie Van Wyk Skavenski[2],
Sergiy Bogdanov[3], Alona Pastukhova[3], Kira Lomakina[3], Paul Bolton[2],
Laura Murray[2] and Judith Bass[2]

[1]School of Education and Human Development, University of Virginia, Charlottesville, VA, USA; [2]Department of Mental Health, Johns Hopkins School of Public Health, Baltimore, MD, USA and [3]Center for Mental Health and Psychosocial Support, National University of Kyiv-Mohyla Academy, Kyiv, Ukraine

## Abstract

Psychosocial programs in low- and middle-income countries (LMIC) often omit cognitive strategies due to perceived difficulty for clients and lay providers. We evaluated the benefit of including "cognitive coping" in a brief, online intervention for conflict-affected Ukrainian veterans and family members with mild to moderate psychosocial distress. Participants were randomized to two treatment conditions based on the Common Elements Treatment Approach Psychosocial Program (CPSS). CPSS-Basic (CPSS-B) included a self-assessment, safety screening and psychoeducation. CPSS-Enhanced (CPSS-E) included these as well as cognitive coping. Distress, functional impairment, alcohol use, aggression, social disconnectedness and conflict resolution were assessed after one month. Participants also evaluated program accessibility, acceptability, appropriateness, feasibility and adoption. Of 1,177 study participants, 788 (67%) completed follow-up. Both conditions significantly improved distress, functional impairment, aggression and social disconnectedness; CPSS-E producing a greater reduction in distress than CPSS-B (ES: $d = .22$, $p = .002$). Implementation outcomes were positive across conditions, favoring CPSS-E for appropriateness ($d = .48$, 95% CI: .33, .62), feasibility ($d = .15$, 95% CI: .00, .29), adoption ($d = .34$, 95% CI: .19, .48) and acceptability ($d = .29$, 95% CI: .15, .44). Findings support the feasibility and added value of incorporating cognitive techniques into psychosocial programming in LMIC.

## Impact statement

This paper describes a trial and related implementation research of a brief, online psychosocial program, delivered by lay providers with limited training, for conflict-affected Ukrainian veterans and family members of veterans. The study compares two active psychosocial interventions that differ only in whether cognitive coping is included. MHPSS interventions commonly implemented in humanitarian settings often rely on behavioral and social approaches, rather than cognitive strategies, due to concerns about the difficulty and scalability of cognitive strategies. Our findings challenge these assumptions, showing that cognitive coping training was feasible, preferable and more effective than the comparison. This study also provides an evidence-based model for a tiered system of care to efficiently allocate resources while extending reach in low-resource settings.

## Introduction

Exposure to short- and long-term crises can have long-lasting impacts on population psychological well-being (Inter-Agency Standing Committee, IASC, 2007). Mental health and psychosocial support (MHPSS) programs can address the psychological impacts of these crises through clinical, psychological and psychosocial services. However, in most contexts where humanitarian crises occur, challenges such as provider shortages and weak service systems often require that MHPSS interventions be delivered by providers with limited or no prior mental health training (Bolton et al., 2023). There is good evidence that MHPSS interventions can be effective when delivered by lay providers when careful consideration of intervention components, local acceptability, training strategies and implementation supports are available to support and sustain interventions at scale (Ryan et al., 2021). This highlights the need for innovations in workforce

training, brief and scalable delivery models and health systems integration (Keynejad et al., 2021; Tol et al., 2023).

Most of the widely used and disseminated brief psychosocial programs use behavioral and social, rather than cognitive, techniques to support reductions in psychological distress and improve social functioning. For example, Problem Management Plus (PM+), (Mwangala et al., 2024), incorporates elements of stress management, problem solving, behavioral activation and social support; these elements were identified by international experts as likely easier to learn by lay workers than cognitive techniques (Dawson et al., 2015). Likewise, Self-Help Plus (SH+; Epping-Jordan et al., 2016), teaches mindfulness-based approaches to refocus from difficult thoughts to the present moment (e.g., grounding, unhooking), being kind to self and others, and values-based problem-solving (World Health Organization, 2021). Both PM+ and SH+ can be delivered and facilitated by lay providers (Epping-Jordan et al., 2016) and are supported by the World Health Organization for their scalability. A recent scoping review found that while PM+ has moderate to large impacts on reducing general distress in the short term, long-term effects were not found and lower effect sizes were found for posttraumatic stress symptoms compared with general distress and depression/anxiety symptoms (Mwangala et al., 2024).

The Common Elements Treatment Approach (CETA) is a transdiagnostic psychotherapy system of care that uses cognitive and behavioral elements and can be delivered effectively by a range of different non-specialist providers (Murray et al., 2014a). There is a strong evidence-base for CETA as an 8–12 session individual psychological treatment for common mental and behavioral health conditions, including depression, anxiety, posttraumatic stress, alcohol use and interpersonal violence (Bolton et al., 2014; Weiss et al., 2015; Murray et al., 2020). Given this evidence, our team has been working in Ukraine since 2015 to strengthen community-based mental health service delivery for people affected by the ongoing conflict. To support a broader stepped care approach, we developed a briefer 5-session version of CETA. Results from a trial comparing brief CETA to full CETA and a no-treatment control showed that both full and brief individual CETA were successful in reducing depression, anxiety and posttraumatic stress symptoms and dysfunction among conflict-affected Ukrainians, with higher effect sizes for full CETA (Bogdanov et al., 2021).

We also observed high support needs yet lower levels of help-seeking and treatment engagement among veterans during the CETA trial. As such, when subsequently scaling up CETA, we further developed and piloted a "light-touch" CETA Psychosocial Support (CPSS) intervention that could be integrated into existing community-level health and social service systems, serving as both a preventive psychoeducational intervention and an accessible entry point for screening and referral within a broader system of care. A comprehensive description of the CPSS development process has been published previously (Nguyen et al., 2023). Briefly, CPSS is a modifiable, single or 2–3 session program designed to be delivered by non-specialist providers with approximately 2–4 days of training. Depending on the elements selected, the CPSS intervention may be delivered in a group or individual format with sessions lasting as brief as 20 minutes up to 2 hours. Standard CPSS elements include mental health screening, psychoeducation, safety assessment and referral as needed. Together, these components comprise CPSS-Basic (CPSS-B) and are the minimum elements to be included. The CPSS program can then be enhanced (CPSS-E) by integrating one or more additional CETA elements based on priority psychological and psychosocial needs of the target population. In Ukraine, CPSS-E included cognitive coping (see below) based on the extensive evidence that cognitive coping is a helpful preventative tool for the general population (Frydenberg et al., 2004; van Starrenburg et al., 2016; Nakao et al., 2021) as well as feedback from participants and providers during the prior CETA trial.

## Specific objectives

To evaluate the feasibility and added value of including a cognitive technique in a brief, group psychosocial intervention, we compared the effectiveness and implementation of CPSS-B workshops with CPSS-E workshops that included cognitive coping training. All workshops were delivered virtually by native Ukrainians, with workshop content developed for Ukrainians and tailored to veterans' needs and experiences. Psychosocial outcomes included general distress, functional impairment, alcohol use, aggression, social disconnectedness and positive conflict resolution. Implementation outcomes included participant perspectives on accessibility, acceptability, feasibility and adoption. We also assessed uptake of referral to full CETA treatment among participants with continued elevated symptoms at follow-up.

## Methods

### Study setting and design

This two-armed, cluster-randomized trial was conducted with Ukrainian veterans and adult family members of veterans between September 2020 and February 2022 – prior to the Russian invasion of Ukraine. Both arms used online delivery of CPSS workshops facilitated by trained CPSS providers. Within the provider, workshops were randomly assigned on a 1:1 ratio to either CPSS-B or CPSS-E. Psychosocial outcomes (Aim 1) were assessed at baseline and 1-month post-workshop follow-up; at follow-up, participants also completed an implementation survey (Aim 2) and were referred to CETA if eligible (Aim 3). Ethical approval was granted both by the Kyiv-Mohyla Academy Ethical Review Committee and the Institutional Review Board at Johns Hopkins School of Public Health (#9782). The trial protocol was registered on clinicaltrials.gov (NCT04234815).

### Interventions and implementation

#### Interventions

CPSS-B and CPSS-E were both delivered as single-session online workshops to groups of 5–10 participants. A comprehensive description of CPSS – including the iterative development and co-creation process in Ukraine – is available elsewhere (Nguyen et al., 2023). Briefly, the program drew from CETA materials previously adapted and tested for Ukraine, were co-developed by English and Ukrainian team members, and included:

1) *Screening*: self-assessment using a brief, validated mental health assessment (see outcome measures).
2) *Psychoeducation*: instruction about the brain and brain functioning; normalization of psychological responses to stress and review of self-assessment results, including what indicates a potential need for additional support.
3) *Safety*: screening, support and referral for individuals at risk of harm to self or others (Murray et al., 2014b). Screening was built

into the self-assessment with two questions assessing thoughts of harm to self or others over the past month. Any participant who indicated a potential safety risk as evidenced by a score of 1 or higher (i.e., any response other than "none") on either question was contacted by a provider within 24-hour to conduct a comprehensive safety assessment. Further supportive actions, such as safety plans, treatment referrals or emergency responses, were initiated according to a standardized safety protocol. Additional safety content was also included during the workshop as part of the psychoeducation regarding normalization of stress responses and identification of additional support needs.

CPSS-E included the additional CETA element of cognitive coping, referred to in the CETA program as "Thinking a Different Way – Part 1" (TDW1). Research has consistently found that cognitive coping, a key component of Cognitive Behavioral Therapy (CBT), is an effective prevention tool across various mental health conditions, with studies showing that by teaching individuals to identify and challenge negative thought patterns, they can proactively manage stress, anxiety and potential triggers for relapse (Frydenberg et al., 2004; van Starrenburg et al., 2016; Nakao et al., 2021). In cognitive coping, participants practice identifying unhelpful thoughts related to a wide range of situations, changing them to more helpful alternatives, and examining how these thoughts impact their feelings and behaviors. In CPSS-E, the providers first used a neutral example (one that they provided, such as getting caught in the rain) to help participants see the connections between thoughts, feelings and behaviors. Participants were then invited to suggest other examples from their own lives (neutral and stressful) to work through together. Finally, there was time to allow each participant to apply the technique to a personal example, with support from the providers. Whereas the CPSS-B workshop was roughly 1 hour in length, the cognitive coping element was delivered after the psychoeducation and safety components, adding approximately 30 minutes to the total workshop time.

### CPSS providers

Providers were native Ukrainians recruited from community-based organizations that were currently providing social services to veterans and/or their families. There were no specific minimum education or training criteria to be selected as a CPSS provider. Interested organizations were invited to nominate providers, who were then interviewed prior to being accepted for training. A majority of the 41 providers (26 female, 15 male) had completed university education, 19 were veterans or military volunteers, 6 were direct family members of veterans, and the remainder had strong connections to the veteran community. Four experienced Ukrainian CETA providers who were trained in CPSS during the development process were mentored to become CPSS trainers; they led the CPSS-B and CPSS-E trainings and provided weekly supervision during the study, working closely with the CPSS development team.

Providers completed a 2-day practice-based training in all the components of the CPSS workshops, group facilitation and safety response. Given the integration of research and program components, providers completed an additional 1-day training in study methods and research ethics, including training on the two CPSS-B and CPSS-E workshop programs. Following training, providers participated in practice groups in which they facilitated both CPSS-B and CPSS-E workshops with fellow trainees as workshop participants, under supervisor observation. All providers delivered both CPSS-B and CPSS-E versions of the workshop (see randomization below) and received ongoing clinical supervision, which included discussion of challenges and collaborative review of provider-reported fidelity checklists; supervisors also periodically observed workshops.

### CPSS online implementation

CPSS was originally developed and piloted as an in-person workshop (Nguyen et al., 2023). Due to the COVID-19 pandemic, the workshops were converted to an online format delivered over Zoom, typically by providers working in pairs. Implementation adjustments for this trial included developing an online registration and survey system administered via Qualtrics (Provo, UT) that enabled participation country wide. After completing a brief registration form with contact information, interested attendees were directed to a separate online form linked by ID number to complete the self-assessment; review of self-assessment results was covered during the workshop. After completing the self-assessment, attendees selected an upcoming workshop (multiple days and time options were provided until workshops were full) and were then provided meeting confirmation, login information and written guidance for joining that included considerations for protecting safety and privacy online. A CPSS provider reviewed all new registrations and assessments daily to identify anyone needing a safety follow-up, which was provided within 24-hour.

Additional adjustments to support online delivery included developing slides for visual engagement, as well as including steps in the training and CPSS facilitator guide to ensure attendees' safety and engagement in an online meeting room (e.g., making sure people were joining from a private space, discussing whether or not to share video). A research assistant was available at the beginning of each workshop to provide technical support to attendees who were struggling with technical issues. Each session began with a welcome, introductions and agenda setting with core topics and ground rules (including the safety considerations noted above), followed by an introduction to stress and then a review of the self-assessment results and discussion of safety. In CPSS-E, the group then moved into didactic and experiential cognitive coping. Both groups ended with a wrap-up with next steps that included a review and encouragement to continue using the information at home, guidance on where to seek additional help if needed, and thanks for participation. A majority of workshops were delivered in Ukrainian, though to maximize accessibility, some bilingual providers offered workshops in Russian for Russian-speaking Ukrainians.

### CPSS follow-up

Within 1 week following the CPSS workshop, providers called each workshop attendee who reported at least minimal distress (self-assessment score $\geq 8$) on the registration self-assessment. For CPSS-B participants, this was simply a brief check-in to discuss any follow-up questions or concerns they had about their self-assessment. For CPSS-E participants, this follow-up call included a review of cognitive coping practice over the previous week, additional instruction to support correct use of the skill, and encouragement to continue practicing. In both conditions, participants were reminded that they would be re-contacted in a month to complete the follow-up self-assessment, at which time anyone with an indicated need (self-assessment score $\geq 13$) would be referred to the full CETA psychotherapy program.

### Participants and recruitment

Given the expectation that both CPSS versions were likely beneficial, and noting that Ukrainian veterans may struggle to overcome barriers to help-seeking, we designed recruitment and consent procedures in consultation with veterans and service providers to support feasibility and acceptability. Advertisements for the workshops were disseminated *via* regular communication channels at collaborating organizations, such as in-person recruitment conversations, paper flyers and regular social media posts. CPSS providers were also incentivized to circulate information about upcoming workshops within their own networks. All Ukrainian veterans and adult family members of veterans were encouraged to attend regardless of perceived need or prior services contact.

### Consent procedures

At the beginning of each workshop, CPSS providers informed participants that the workshop was being evaluated through a research study, but that everyone was welcome regardless of their interest in study involvement and they would share more information about the study at the end of the session. Following completion of all the psychosocial programmatic activities, the CPSS providers gave an overview of the study and read an oral consent script to the entire group, while a study assistant direct messaged an online consent form to each attendee using a Qualtrics form with a unique URL that was linked to their registration ID number. Attendees were asked to complete the form to either confirm or decline study participation; if no response was provided, CPSS providers attempted a second consent during the 1-week follow-up phone call. Study participation was not required for programmatic follow-up supports.

### Inclusion and exclusion criteria

Ukrainian veterans and adult family members of veterans who attended the workshop and who scored an 8 or above on the self-assessment (i.e., indicative of at least some level of distress) were eligible for study inclusion. Participants scoring lower than 8 were excluded from the trial but were still eligible to participate in the programmatic activities. Participants identified with safety concerns requiring immediate referral to higher-level care (e.g., immediate CETA, hospitalization, other institution-based care) were also excluded. Other exclusion criteria included 1) current active duty in the military (i.e., not yet a retired veteran); 2) late arrival to the workshop such that they missed the self-assessment review (i.e., insufficient intervention dose) and either 3) previous CETA participation or 4) being themselves a current MHPSS service provider. Any participant who provided consent at the end of the workshop but was subsequently determined to be ineligible was informed by email that they would not be enrolled in the study but were welcome to reach out to their group leader any time if they felt they would benefit from additional support or consultation.

### Sample size

The power calculation for Aims 1 was conducted using the *Power Up!* (Dong and Maynard, 2013) sample size calculator for a 3-level random effects blocked cluster randomized assignment design with treatment at level 2 to account for the group-level clustering and within-provider randomization for Aim 1. Given the light touch of the intervention and relatively low level of distress required for inclusion, we targeted a sample size that would enable detection of a small effect size (0.30) at 80% power with a two-tailed $p = .05$

significance level. Assumptions include a minimum of 20 provider pairs (i.e., Level 3), offering on average 10 sessions each (i.e., Level 2), from which an average of 3 eligible participants would consent and contribute data. We anticipated minimal clustering but included a more conservative estimate of ICC = 0.1 at Level 2 and ICC = 0.2 at Level 3, as well as the potential for heterogeneity in treatment effect across providers of .5. These assumptions indicated a necessary sample size of $N = 603$ participants for Aim 1, which we inflated by 30% to allow for expected dropout, resulting in a target of $N = 784$ for pre-post analysis. To support Aim 2 (implementation outcomes) and Aim 3 (follow up service uptake among participants with higher need), we obtained IRB approval to recruit up to 1,387 participants. In this scenario, for Aim 3, assuming 43% of participants ($N = 603$) would be referred to treatment and estimating a treatment engagement rate of 40% in the comparison condition, we would be powered to detect an absolute 18% increase in treatment initiation (i.e., 58% vs. 40%).

### Randomization and concealment

Cluster-level randomization to CPSS-E or CPSS-B was carried out at the workshop level within provider pairs. Prior to launching the study, a US-based member of the study team (AJN) generated the randomization scheme. First, we generated a list of Provider and Session numbers that would allow for up to 18 sessions per provider pair. Within each provider pair, sessions were divided into blocks of six. Within each block, the six sessions were allocated on a 1:1 ratio to either CPSS-E or CPSS-B. This blocking strategy was used to ensure balanced allocation regardless of whether some providers offered more sessions than others. This process was repeated to create a total of five randomization schemes, each with a unique set seed for reproducibility. The five schemes were then provided to the Ukraine-based Study Coordinator, who randomly selected one of the five schemes for initial use.

Because providers were involved in outreach to potential workshop participants, it was important that upcoming workshop allocation remain concealed until registration was complete. Providers were only informed of session allocation 24-hour in advance (to allow time to prepare), at which time no further registrations would be processed. To minimize predictability, providers were not informed that randomization was completed in blocks of six workshops.

### Blinding

While the study coordinator and providers were aware of participants' allocation status at time of the workshop delivery, all US-based study team members were blinded for the duration of the study, including through completion of preliminary analysis. Participants were informed that there were two workshop versions and "both versions of the workshop include information and skills that we know are likely to be helpful, but one version teaches an extra skill." It is reasonable to assume that some participants were able to deduce their allocation status.

### Data collection

Baseline data were collected *via* the self-administered online assessment during the registration process, as described above. Follow-up data were collected 1 month after workshop attendance also using the self-administered online assessment as well as an additional implementation outcomes survey. Unique survey links were sent to

participants *via* their preferred phone or email contact. Survey completion was monitored, and up to three reminders sent before treating the participant as lost to follow-up. CETA referrals and treatment initiation were collected from program records.

### Outcome measures

**Primary outcomes** (Aim 1) were assessed using a locally validated instrument used in the prior CETA trial (Doty et al., 2018; Bogdanov et al., 2021). ***Distress*** was assessed with 21 items assessing symptoms of depression, anxiety and post-traumatic stress. Participants reported the frequency of experiencing each item over the past month, with responses on a scale of 0 "none of the time" to 3 "almost all of the time." For assessing inclusion and referral requirements, distress items were summed for a total score (range: 0–63), with a minimum score of 8 (i.e., mild distress) for study eligibility. A score of 13 or higher indicated moderate to severe distress requiring CETA referral at follow-up; these cut-off scores were derived from the previous CETA trial data (Bogdanov et al., 2021). For trial outcome analyses, scores were calculated as the mean of contributing items. ***Functional impairment*** was assessed with eight items derived from the World Health Organization Disability Assessment Schedule (WHODAS; World Health Organization, 2010) and local qualitative research. Items assess the level of difficulty doing daily tasks and activities over the past month relative to same-aged peers, with responses ranging from 0 "no more difficulty" to 4 "so much more difficulty that I often cannot do." Internal consistency for the distress and functional impairment scales was high (α = .92 and α = .86, respectively). Minimal clustering was observed in baseline (ICC = .05 and ICC = .02, respectively) or change scores (ICC = .02 and ICC = .07, respectively).

**Secondary outcomes** (Aim 1) included past-month alcohol use, aggression, social disconnect, and positive conflict resolution. As locally validated tools were not available, we sourced, translated, and piloted items from existing tools that reflected priority concerns raised during formative qualitative work with Ukrainian veterans and their families (Nguyen et al., 2023). ***Alcohol use*** was measured using the AUDIT-C (Bush et al., 1998), a 3-item measure of alcohol consumption. Responses were calculated as a total score ranging from 0 to 12 with higher scores indicating greater alcohol use. Internal consistency was satisfactory (α = .74). ***Aggression*** was measured with four items from the Revised Conflict Tactics Scale (CTS; Newton et al., 2001) assessing verbal, physical, and property-related aggression. Responses reported frequency of engaging in each behavior on a scale of 0 "never" to 3 "4 or more times." Internal consistency was satisfactory (α = .74). ***Social disconnection*** was measured with five negatively worded items (e.g., feel disconnected, misunderstood, do not feel like I belong) from the Social Connectedness Scale – Revised (Lee and Robbins, 1995). Response options ranged from 0 "strongly disagree" to 5 "strongly agree" such that higher scores indicated greater social disconnection. Internal consistency was high (α = .91). ***Positive conflict resolution*** was measured with two items from the CTS (Newton et al., 2001) assessing frequency of compromising and showing respect during disagreement on the same response scale as the aggression items. Responses reported frequency of engaging in each behavior on a scale of 0 "never" to 3 "4 or more times." Correlation between these two items was satisfactory (*r* = .79). For all outcomes, higher scores indicated more psychosocial difficulties, with the exception of positive conflict resolution for which higher scores indicated more frequent use of prosocial conflict resolution strategies.

**Implementation outcomes** (Aim 2) were assessed using the locally validated Mental Health Implementation Science Tools Consumer Version (Haroz et al., 2019; Aldridge et al., 2022). This instrument includes scales measuring participant perspectives on program ***adoption*** (4 items, α = .78), ***acceptability*** (10 items, α = 87), ***appropriateness*** (3 items, α = .84), ***feasibility*** (8 items, α = .74) and ***accessibility*** (8 items, α = .84). Response options ranged from 0 (disagree) to 3 (agree) and scale scores were calculated as the mean of contributing items, with higher mean scores indicating a more positive perspective on that implementation domain. Interclass correlation coefficients (ICCs) ranged from .00 (feasibility) to .05 (appropriateness, accessibility).

**Treatment engagement** (Aim 3) was operationalized as a binary variable indicating whether or not participants who were given a CETA referral at follow-up attended an initial meeting with a CETA counselor within the next 3 months. The original protocol indicated a 1-month follow-up window; however, we observed a variety of pragmatic reasons for which the referral-to-initiation window extended beyond 1 month and so revised this outcome period.

### Statistical analysis

All analyses were conducted in Stata 17.0 (StataCorp, 2021). Internal consistency of each scale was evaluated using Cronbach's alpha. For all outcome analyses, scale scores were calculated as the mean of contributing items (except alcohol use which maintained as total score), to retain the original response scale and to account for skipped or not applicable items. Scales missing more than 40% of contributing items were recoded as missing. Clustering by provider was evaluated at baseline and follow-up and in change scores.

Baseline scores and demographic characteristics were compared between the study arms using multi-level logistic (for binary) and linear (for continuous) regression models with clustering by provider to evaluate the success of the randomization strategy. A binary loss to follow-up (LTFU) indicator was regressed onto baseline variables to identify predictors of LTFU that could impact outcome analyses. For Aim 1, change scores were calculated by subtracting the baseline score from the follow-up score, such that a negative change value indicates improvement (i.e., reduction in psychosocial problem) with the exception of positive conflict resolution, for which a positive change value indicates improvement. Following a difference-of-differences approach, change scores were compared between the study arms using multi-level linear regression models with clustering by provider. Given the minimal clustering observed, we also generated unclustered effect size estimates (Cohen's D) for each outcome. Aims 2 and 3 used the same analytic approach with multi-level linear and logistic regression models, respectively.

### Sensitivity analyses

During the trial, one provider pair was noted multiple times as being different in meaningful ways, such as disproportionately high recruitment rates, vocal preference for CPSS-E relative to CPSS-B, and fidelity concerns identified in supervision (e.g., deviating from the standard script in response to the needs of the group). To examine potential provider heterogeneity in results, we re-ran the analysis for the primary distress outcome with iterative removal

and replacement of each provider pair. As this analysis identified the problematic pair as a notable outlier, all analyses were re-conducted excluding all data from this pair. Both the full data and the sensitivity analysis results are presented below.

## Results

### Participants and study flow

In total, 2,548 people completed an online registration and the baseline assessment during the study recruitment between September 2020 and December 2021. Of these, 1,631 (64%) attended a workshop; approximately 89% of attendees (1,453) were eligible for the study and 81% of those eligible (1177) consented to participate. Supplemental Table S1 provides sample descriptions at each phase of the study relative to all program registration data. The consenting study sample is largely reflective of those who initially registered, with the exception that the study sample has more family members than veterans (51.8%

vs. 48.2%), whereas more veterans than family members were initially interested and completed baseline assessments (54.2% vs. 45.8%). Participants who reported being referred by providers appeared more likely to attend and consent than those who reported learning about the workshops from social media advertisements.

The CONSORT diagram (Figure 1) shows that the two study arms were similar in size (50.7% of study participants in CPSS-B and 49.3% in CPSS-E). The consent rate was significantly higher among CPSS-B participants relative to CPSS-E (84.9% vs. 77.4%, *p* < .001); however, the two groups were statistically comparable on all baseline indicators (Table 1).

**Loss to follow-up (LTFU)** between the CPSS workshops and the 1-month follow-up assessment was approximately 33% and was similar between groups (33.2% in CPSS-B, 32.9% in CPSS-E, *p* = .890). Adjusting for gender, veteran status, and baseline scores, two indicators were predictive of LTFU: veterans had 41% lower odds of LTFU than family members (OR = 0.59, *p* = .016), and participants with higher feelings of social disconnectedness had

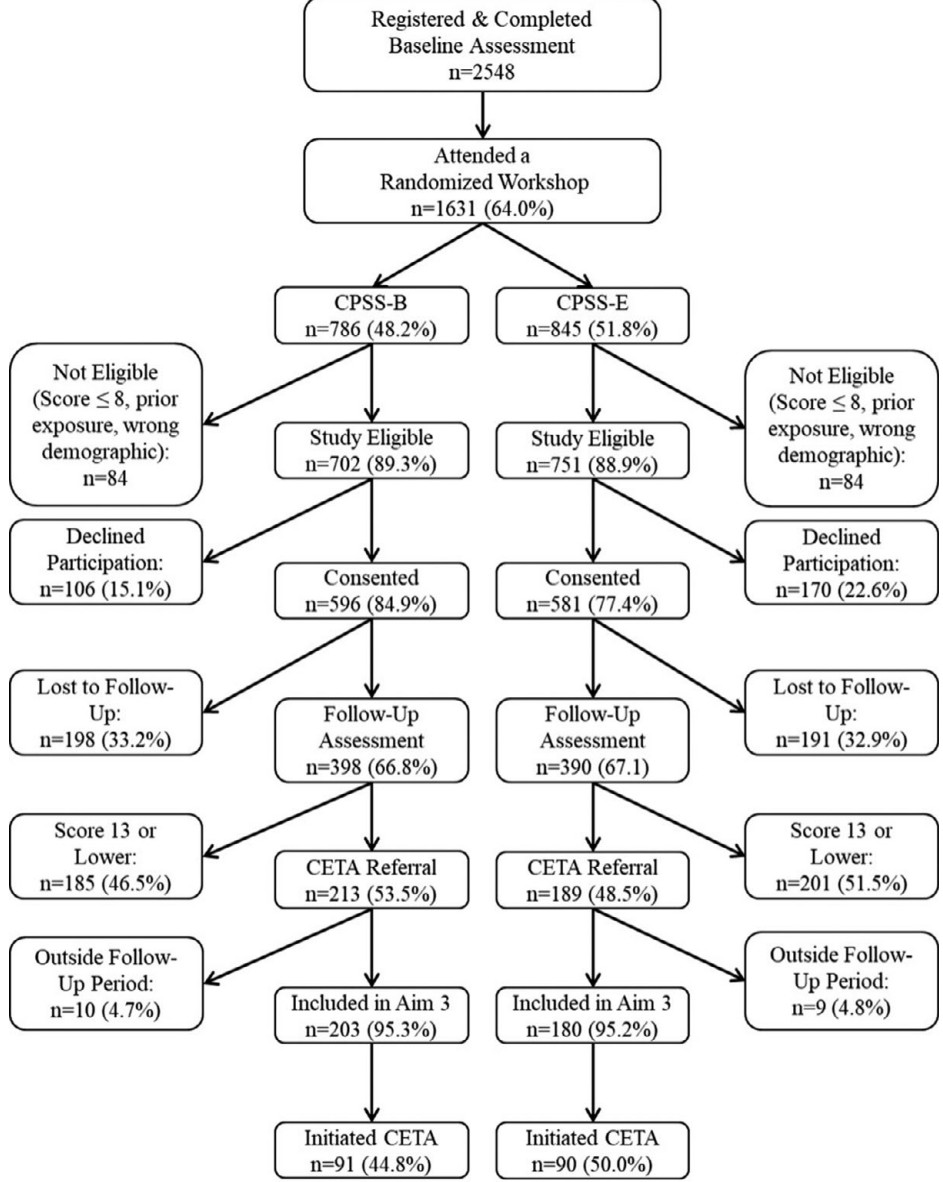

**Figure 1.** Consort diagram.

**Table 1.** Baseline comparisons by allocation (*n* = 1,777)

| | CPSS-B (*n* = 596) %/Mean (SD) | CPSS-E (*n* = 581) %/Mean (SD) | *p*-value[a] | Adjusted *p*-value[b] |
|---|---|---|---|---|
| Veteran (vs. family member) | 50.7% | 45.6% | .082 | .124 |
| Male (vs. female) | 43.2% | 38.9% | .134 | .148 |
| Age | 40.5 (10.1) | 40.9 (11.2) | .499 | .449 |
| Eastern region | 36.3% | 36.1% | .937 | .550 |
| Distress | .96 (.42) | .99 (.45) | .221 | .216 |
| Functional impairment | .81 (.58) | .85 (.63) | .339 | .406 |
| Alcohol use | 1.94 (1.94) | 2.06 (2.11) | .350 | .248 |
| Aggression | .56 (.62) | .56 (.61) | .959 | .798 |
| Social disconnectedness | 1.75 (1.06) | 1.82 (1.06) | .251 | .457 |
| Positive conflict resolution | 1.59 (.87) | 1.65 (.87) | .250 | .168 |

[a]Chi-square test for binary, *t*-test for continuous variables.
[b]Using multi-level logistic/linear regression with clustering by provider.

39% higher odds of LTFU compared with those with less social disconnectedness (OR = 1.39, *p* = .002).

### CPSS-E versus CPSS-B program effects (Aim 1)

Within-group change and between-group comparisons in psychosocial outcomes are reported in Table 2. Clustering of outcome scores by provider pair was negligible, with ICCs ranging from .06 (alcohol use) to .10 (social disconnect) at baseline and .02 (alcohol use, positive conflict resolution) to .05 (aggression) in change scores. Note that while the overall follow up sample was *n* = 788, the actual analytic sample varied by outcome due to item-level missingness within scales (see Table 2). Significant within-group improvements were observed for distress, functional impairment, aggression, and social disconnectedness in both arms. Across study arms, only distress was significantly different, favoring CPSS-E participation ($\gamma = -.09$, SE = .03, *p* = 0.002). This magnitude of difference approximates a modest effect size of *d* = .22 (95% CI: .08, .36).

### Implementation outcomes (Aim 2)

At follow-up, 759 participants (96.3% of all participants providing follow-up data) completed the implementation survey. Table 3 shows ratings were largely positive for all domains across both arms, with means ranging from 2.19 (Appropriateness) to 2.68 (Feasibility) for CPSS-B and 2.50 (Adoption) to 2.75 (Acceptability) for CPSS-E. Statistically significant differences favoring CPSS-E reflect a moderate effect size for appropriateness (*d* = .48, 95% CI: .33, .62), small effect sizes for feasibility (*d* = .15, 95% CI: .00, .29) and modest for adoption (*d* = .34, 95% CI: .19, .48) and acceptability (*d* = .29, 95% CI: .15, .44).

### CETA referrals and uptake (Aim 3)

Of the 788 participants who completed follow-up assessments, 383 (48.6%) scored 13 or higher, indicating at least moderate distress, and were referred to full CETA. A non-significant difference was observed between CETA eligibility rates in the two conditions (CPSS-B: 53.5% CPSS-E: 48.5%; OR = 0.82, 95% CI: .60, 1.10; *p* = .185). Of these, 91 out of 203 CPSS-B participants (44.8%) and 90 out of 180 CPSS-E participants (50%) initiated full CETA. This difference favored CPSS-E but was not statistically significant (OR = 1.21, 95% CI: .77, 1.90; *p* = .404).

### Sensitivity analysis

Estimated effect sizes for the distress outcome with iterative removal and replacement of each provider pair are plotted in Supplemental Figure S1, illustrating the disproportionate influence of the one provider pair that was identified as potentially problematic during study implementation; this pair accounted for 15.5% (*n* = 245) of the full study sample (see Supplemental Figure S2 for a revised CONSORT diagram with this pair's sample removed). Removing all data from the participants associated with these providers did not change statistical inferences for any of the psychosocial outcomes or implementation outcomes, although effect sizes increased slightly in favor of CPSS-E (Table 4). The statistically significant $\gamma = .12$-point greater decrease in distress among CPSS-E participants relative to CPSS-B approximates an effect size of *d* = .29. Though not statistically significant, odds of meeting referral cutoff were estimated at 27% lower in CPSS-E than CPSS-B (56.1% vs. 63.5%; OR: .73, 95% CI: .52, 1.03; *p* = .07), and odds of

**Table 2.** Within-group change and between-group difference-of-differences in psychosocial outcomes (ITT, *n* = 788)

| | | CPSS-B | | | | CPSS-E | | | Difference of differences | | Effect size estimate | |
|---|---|---|---|---|---|---|---|---|---|---|---|---|
| | *n* | Pre mean (SD) | Post mean (SD) | Change mean (SD) | *n* | Pre mean (SD) | Post mean (SD) | Change mean (SD) | γ (SE) | *p*-value | *d* | 95% CI |
| Distress | 396 | .94 (.40) | .71(.43) | −.23 (.39)[a] | 389 | .99 (.47) | .67 (.45) | −.32 (.42)[a] | −.09 (.03) | .002 | .22 | .08, .36 |
| Functional impairment | 388 | .83 (.56) | .58 (.52) | −.25 (.54)[a] | 372 | .85 (.61) | .60 (.61) | −.25 (.55)[a] | .01 (.04) | .726 | −.01 | −.15, .13 |
| Alcohol use | 333 | 2.01 (1.85) | 1.90 (1.88) | −.11 (1.60) | 326 | 2.18 (2.09) | 2.02 (2.07) | −.16 (1.63) | −.07 (.13) | .601 | .03 | −.12, .19 |
| Aggression | 341 | .51 (.61) | .39 (.46) | −.12 (.61)[a] | 325 | .55 (.61) | .42 (.51) | −.13 (.58)[a] | −.02 (.04) | .601 | .02 | −.13, .17 |
| Social disconnectedness | 390 | 1.73 (1.05) | 1.45 (1.04) | −.28 (.91)[a] | 384 | 1.83 (1.07) | 1.44 (1.10) | −.40 (.84)[a] | −.11 (.06) | .085 | .13 | −.01, 28 |
| Positive conflict resolution | 352 | 1.57 (.87) | 1.61 (.89) | .04 (.91) | 343 | 1.63 (.86) | 1.63 (.91) | .00 (.99) | −.05 (.07) | .526 | .04 | −.11, .19 |

[a]Significant (*p* < .05) within-group (pre-post) change.

**Table 3.** Between-group differences in participant implementation perspectives at follow-up (ITT; *n* = 759)

|  | CPSS-B | CPSS-E | Between-group comparison | | | |
|---|---|---|---|---|---|---|
|  | Mean (SD) | Mean (SD) | *γ* (SE) | *p*-value | d | 95% CI |
| Adoption | 2.27 (.80) | 2.51 (.62) | .23 (.05) | <.001 | .34 | .19, .48 |
| Acceptability | 2.63 (.51) | 2.75 (.33) | .12 (.03) | <.001 | .29 | .15, .44 |
| Appropriateness | 2.19 (.81) | 2.52 (.57) | .32 (05) | <.001 | .48 | .33, .62 |
| Feasibility | 2.68 (.38) | 2.73 (.29) | .05 (.02) | .039 | .15 | .00, .29 |
| Accessibility | 2.53 (.52) | 2.53 (.45) | .01 (.03) | .755 | .01 | −.13, .16 |

**Table 4.** Between-group comparisons with provider pair removed (sensitivity analysis; *n* = 590)

|  |  | CPSS-B | CPSS-E | Between-group comparison | | | |
|---|---|---|---|---|---|---|---|
|  | *n* | Mean (SD) | Mean (SD) | *γ* (SE) | *p*-value | *d* | 95% CI |
| Pre-post change in: | | | | | | | |
| Distress | 587 | −.23 (.41)[a] | −.35 (.44)[a] | −.12 (.03) | <001 | .29 | .13, .45 |
| Functional impairment | 562 | −.17 (.55)[a] | −.22 (.56)[a] | −.04 (.05) | .384 | .08 | −.09, .25 |
| Alcohol use | 470 | −.21 (1.68) | −.33 (1.50)[a] | −.12 (.15) | .407 | .08 | −.10, .26 |
| Aggression | 477 | −.21 (.64)[a] | −.20 (.56)[a] | .00 (.05) | .928 | −.01 | −.19, .17 |
| Social disconnectedness | 576 | −.31 (.97)[a] | −.45 (.85)[a] | −.13 (.08) | .097 | .15 | −.01, .32 |
| Positive conflict resolution | 505 | .01 (.93) | −.02 (1.01) | −.03 (.09) | .726 | .02 | −.15, .20 |
| Implementation ratings | | | | | | | |
| Adoption | 554 | 2.23 (.82) | 2.47 (.63) | .24 (.06) | <.001 | .33 | .16, .50 |
| Acceptability | 558 | 2.59 (.52) | 2.73 (.33) | .14 (.04) | <.001 | .32 | .15, .49 |
| Appropriateness | 553 | 2.13 (.87) | 2.50 (.57) | .36 (.06) | <.001 | .50 | .33, .67 |
| Feasibility | 556 | 2.67 (.41) | 2.72 (.31) | .06 (.03) | .065 | .16 | −.01, .32 |
| Accessibility | 552 | 2.50 (.56) | 2.50 (.47) | .01 (.04) | .844 | 0.00 | −.17, .17 |

[a]Significant (*p* < .05) within-group (pre-post) change.

initiating CETA were estimated at 39% higher in CPSS-E than CPSS-B (54.5% vs. 45.6%; OR: 1.39, 95% CI: .86, 2.26, *p* = .178).

## Discussion

This study compared two versions of a brief, online group psychosocial workshop that varied only by one component: presence or absence of cognitive coping skill building. One-month post-workshop, participants in both arms showed small improvements in a range of psychosocial problems and reported positive perceptions of both workshops. Relative to the workshop without cognitive coping (CPSS-B), inclusion of the cognitive coping training activity (CPSS-E) resulted in significantly greater reduction in distress and higher participant ratings regarding likely adoption, acceptability, and perceived appropriateness.

Another aim of the study was to understand the level of need for full CETA within a prevention group and to explore the uptake of full CETA among those who were referred. Although the magnitude of the difference in distress reduction between the two arms was small, it *could* result in a practical difference in the proportion of participants requiring full CETA – while underpowered, our results are promising enough to support further study with a larger

referral sample. Particularly in areas where mental health service demand is high and mental health provider availability is limited, this type of low-intensity, scalable intervention may be useful as a first-tier service for people who are struggling to cope with everyday stressors but who may not require ongoing individual support, freeing up provider time for higher-need cases (Bolton et al., 2023).

While the potential implications for distress reduction are important, participants' perspectives on implementation outcomes are equally pertinent. Relative to CPSS-B, participants who received CPSS-E reported *liking* the workshop more and perceived the workshop content as being a better fit to their needs. The validity of these differences is reinforced by the relative *lack* of differences between the two groups in perceived accessibility; whereas the content differences conceptually account for differences in perceptions of usefulness and fit, there would be no reason to expect accessibility be different.

This study set out to address the question of whether including a therapeutic cognitive technique, specifically cognitive coping skills, was useful and feasible in a low-intensity psychosocial intervention. The quantitative findings supporting inclusion of cognitive coping are reinforced by anecdotal provider experiences throughout the trial, in which providers described feeling as though CPSS-B did not have as much to offer and was not as well-received by participants

as compared with the CPSS-E workshop. The study results provide initial evidence that inclusion of cognitive coping has at least a short-term advantage over basic psychosocial programming with screening and psychoeducation for both symptom reduction and end-user preferences. Importantly, we also demonstrated that non-specialist psychosocial providers can be effectively trained and supervised to deliver cognitive coping skills training with fidelity in this brief, online, group format. Our findings challenge assumptions about the difficulty teaching and scaling cognitive intervention components, and provide a model for training and design of effective, scalable, tiered system of care in low-resource settings.

Finally, part of the motivation for developing this brief psychosocial workshop was to have a brief intervention that would serve both as a psychosocial support program as well as an identification and pathway to care for those needing more extensive mental health services. It is notable that in this prevention program, a large proportion of attendees did have substantial symptoms meriting referral. This highlights the need for a system of care with multiple tiers of supports available, and continuous monitoring of needs, referral rates, and rates of uptake.

### Study limitations

This study compared two versions of an active intervention that were both expected to provide therapeutic benefit. Lacking a standard control condition, we are unable to make broader statements about the overall effectiveness of the CPSS program, instead restricting our interpretation to the added value of cognitive coping. However, with consensus recommendations supporting wide provision of basic psychosocial supports (IASC, 2007), we did not feel restricting access to any basic intervention would be useful or ethical. The comparisons we chose also limit what conclusions we can draw about the relative value of cognitive coping; while findings support its added value beyond basic screening and psychoeducation, we cannot conclude it would be more or less effective or preferable than other active training elements. However, this study does provide evidence for the feasibility and initial benefits of including cognitive techniques in basic psychosocial programming.

Beyond the lack of a no-intervention control condition, this type of pragmatic, hybrid implementation-effectiveness study in real-world conditions introduces a number of other challenges that must be recognized. Most critically, prioritizing accessibility and a common sense program flow over study design resulted in a participant consent process that occurred after intervention participation. That consent rates subsequently varied across the two arms raises the challenge of interpretability. Although the study groups were similar on measured baseline indicators, it is possible that results reflect some level of selection bias; for example, if participants who disliked CPSS-E were less likely to consent to study participation, this could explain the difference in consent rates as well as the apparent favorability of CPSS-E implementation ratings. Similar questions are raised by the higher rate of consent among family members than veterans, which we believe to some extent reflects that many family members attended out of motivation to help their loved ones.

We also highlighted challenges in maintaining participant and provider blinding with this study design. It is plausible that participants' awareness of their intervention condition could have influenced their self-report responses in ways that either sought to please the study team or to illustrate dissatisfaction with a seemingly "incomplete" intervention.

Loss to follow-up (LTFU) was not random. Participants who felt more socially disconnected – some of those we most wanted to engage – were less likely to participate in the one-month follow-up. While not a surprising finding, this reinforces the need for a variety of strategies to support and engage people with diverse presenting needs.

Finally, we identified some important challenges with one of the provider-pairs in the study. The problems were initially identified during ongoing supervision – in which supervisors periodically joined sessions and regularly reviewed providers' fidelity reports – and also manifest clearly in the sensitivity analyses. This highlights both the necessity and limits of ongoing supervision, particularly in a trial in which providers are delivering both interventions conditions. In this study, we were not able to quickly correct the problem even with trainer involvement but rather ended up removing a substantial amount of data in sensitivity analyses. That the data removal increased effect sizes is consistent with our programmatic inquiry that suggested the fidelity problem was not with CPSS-E but rather that this provider pair was not properly delivering CPSS-B, due to the provider preferences described above. This experience illustrates the critical need for ongoing training, supervision, and fidelity monitoring in MHPSS programming.

### Conclusion

This study describes a comparative evaluation of a brief, single-session online psychosocial workshop that includes screening, psychoeducation, safety and referral, with and without a cognitive coping skill training activity. Although some psychosocial interventions have favored behavioral or social components due to reservations about the difficulty of training and implementing cognitive components (Dawson et al., 2015), this study supports the feasibility of teaching cognitive techniques to both lay providers and clients, with promising benefits both in terms of symptom reduction and participant acceptability and appropriateness perspectives. With nearly half of participants requiring further referral, this real-world research also highlights the critical need to develop effective, multi-tiered programming that can be successfully implemented – and monitored – in challenging and rapidly changing real-world settings. Future research to examine longer-term outcomes such as impact on ongoing clinical care (e.g., need for fewer sessions, or more rapid uptake of coping skills), as well as differential impacts in subgroups or other populations, will be useful to further examine the broader utility of this intervention.

The conclusion of this study coincided with growing apprehension among Ukrainians about the threat of a Russian invasion, which became a reality in February 2022. That invasion left individuals trained for this project, as well as program participants, displaced and living in fear. Some CETA and CPSS providers, as well as participants, went on to serve on the frontlines or as volunteers in the humanitarian response. In the intervening time, we have continued to work closely with available CETA and CPSS providers, with many reporting they continue to use CETA and CPSS skills to provide support to themselves and those around them.

**Open peer review.** To view the open peer review materials for this article, please visit http://doi.org/10.1017/gmh.2025.10065.

**Supplementary material.** The supplementary material for this article can be found at http://doi.org/10.1017/gmh.2025.10065.

**Data availability statement.** Data and intervention materials are available on request.

**Acknowledgments.** We thank the National University of Kyiv Mohyla Academy and the Ministry of Veterans Affairs in Ukraine for their support of this project, and our team of dedicated providers for their tireless work to improve the lives of conflict-affected people in Ukraine.

**Author contribution.** PB, LM, JB, SB and AJN were involved in conceptualization, methodology, project administration and supervision. TR and SB oversaw local project administration. SS, KL, SB and LM provided clinical oversight. AJN, TR, AP and SB led investigation and data collection. AJN, TR and JB collaborated on data analysis and writing, and all authors contributed to review and editing. All authors reviewed and approved the final article.

**Financial support.** This work was supported by the USAID Victims of Torture Fund (contract number AID-OAA-LA-15-00003).

**Competing interests.** The authors declare no conflicts of interest.

**Clinical trials registration.** The trial protocol was registered on clinicaltrials.gov (NCT04234815).

**Ethics statement.** Ethical approval was granted by the Kyiv-Mohyla Academy Ethical Review Committee and the Institutional Review Board at Johns Hopkins School of Public Health (#9782).

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
