## [Reviewer Report]

Thank you for giving me this opportunity to review this manuscript, which provides valuable field data for the field of mental health interventions, particularly with regard to psychological distress in conflict-affected populations, such as veterans and their family members. The study was well-designed, evaluated different versions of an online psychosocial intervention (CPSS-B vs. CPSS-E) through a randomized controlled trial (RCT), and provided valid outcome analyses across multiple dimensions of psychological distress, functional impairment, and more. However, there are still some areas for improvement in the article:

1. The introduction section could have shed more light on the prevailing challenges of mental health interventions in low- and middle-income countries (LMICs), especially in conflict and disaster contexts. The need for the study was reinforced by including more literature on psychological distress in conflict contexts.

2. Compare and contrast in more detail the development process of the CPSS (including the CPSS-B vs. the CPSS-E) and its application in other contexts, clarifying its strengths over other common interventions and helping readers to understand why the study is innovative.

3. Suggests that the Methods section provide more specific implementation details about the CPSS-B vs. CPSS-E intervention components, especially the steps of the online intervention and how participants will engage in the interaction. Specify how to ensure participants' sense of security and privacy protection.

4. Provide detailed information on how to control for bias and confounding variables that may affect outcomes. Further details could be provided on how potential sample selection bias can be controlled for through randomization and analysis strategies.

5. In the results section, the potential impact of attrition on the results could be analyzed in detail, with a discussion of the causes of attrition and differences in the characteristics of the attritors. If possible, a comparative analysis could be conducted to demonstrate whether attrition rates affected the representativeness of the final sample.

6. The benefits of the individual measures, particularly the benefits of cognitive coping skills (CPSS-E) in terms of reduction in psychological distress, could be further analyzed for clinical significance rather than just statistical significance, to help understand the value of this intervention in practical application.

7. Further explicit discussion of the limitations of the study, particularly the high attrition rate and the lack of a control group, and suggestions for how these aspects could be improved in future studies. For example, suggest the inclusion of a control group and discuss the potential for attrition to bias the results.

8. In the conclusion, clearly indicate the direction of future research, especially long-term follow-up and evaluation of effects in different populations, which can provide direction for subsequent academic research and practical application.

---

## [Reviewer Report]

Overall, this article will greatly contribute to the pool of mental health data from Ukraine, particularly around evidence to add cognitive coping to MHPSS support for veterans - as this will be a growing need in the years to come. With a large study sample, the study interventions are quite unique and results that demonstrated reductions in distress are impressive. Kindly see below a few remarks and questions that aim to clarify and strengthen the manuscript:

1. Spell out WHO first time for reader clarity.

2. In the background, I suggest adding that PM+ and SH+ are also preferred to their scalability. This could be integrated into the discussion as the CPSS-E intervention can also be scaled for non professionals.

3. In the background, I recommend reframing symptoms of posttraumatic stress and other MH conditions to “signs” or challenges"

4. Throughout the paper, and particularly in the background, I recommend including more information about the participants and context in Ukraine. This is a unique group and it would be interesting to understand why this group was targeted for this study, and the results could inform future MHPSS programming in Ukraine and beyond.

5. In the objectives and methods section, please add additional details on the workshops and cognitive coping training - were they contextualized for Ukraine? In which language were they delivered? Did this happen through translation?

6. How long was the single online workshop (inclusive of the 30 min cognitive coping session)? Was the 2-day practice training also online?

7. How specifically were CPSS providers recruited?

8. In the discussion section, I believe more information can be added to strengthen the compelling results and how this could inform future programs. For example, why were there more veterans than family members were initially interested than represented in the study sample? How does the supervision component support with fidelity of providers and overall outcome of intervention/study results? Can you add some more narrative to the “anecdotal provider experiences” such as quotations etc.?

9. Finally, the challenge with the one provider-pair seems like it could use more discussion, particularly how it connects to the importance of supervision. It might be useful to add additional details on this point.

---

## [Reviewer Report]

Thank you for the opportunity to review this important and well-written paper.

IMPACT STATEMENT

The current statement is clear but it could be expanded to convey the study’s significance for real-world MHPSS (Mental Health and Psychosocial Support) programming. For example, challenging the assumption that cognitive strategies are too complex for lay providers. The findings also have relevance for policy and practice, as the study provides an evidence-based model for a tiered system of care, where a low-cost, scalable intervention can extend reach, highlighting the potential for efficient resource allocation in low-resource settings.

INTRODUCTION:

The introduction is clear. Only one point: You mention a successful trial of a 5-session “brief CETA.” Before introducing the “light-touch” CPSS (1-3 sessions). Could the rationale for developing an even shorter version be stated more directly?

METHODS

The authors have provided a detailed and transparent methods section that aligns with best practices in trial reporting. The inclusion of a pre-registered protocol, a clear description of provider training, and robust procedures for randomisation and allocation concealment are noted strength. The following suggestions are offered to further enhance the clarity, rigour, and replicability of the study:

1. The Intervention Materials: The manuscript refers to core intervention materials, including the “CPSS facilitator guide,” “slides for visual engagement,” and a self-assessment tool. In line with TIDieR guidelines, the paper could be strengthened by providing access to these materials, if at all possible. This could be either as supplementary files or via a public repository, so that future researchers can more readily replicate the intervention or adapt it to a given context.

2. Intervention Dose: While the CPSS-E workshop is noted to be approximately 30 minutes longer than the CPSS-B workshop, it would be helpful to also state the planned total duration of the CPSS-B session. This would allow for more precise comparison between arms and support reproducibility.

3. Fidelity: The sensitivity analysis appropriately identifies a provider pair with fidelity concerns. To further strengthen transparency, the authors could consider describing fidelity across all provider pairs. For example, via adherence rates from supervision checklists or a qualitative summary of overall delivery. This would help reassure readers about the consistency of intervention delivery across the trial.

4. Acknowledge Blinding Limitations in the Discussion: The methods section notes that participants were likely able to infer their allocation. While full blinding is always challenging in trials, it may still be useful to acknowledge this limitation in the Discussion and briefly consider how it may have influenced self-reported outcomes.

RESULTS

The Results section is transparently reported and is technically precise, providing a clear account of participant flow and appropriately reporting effect sizes and confidence intervals alongside statistical significance. The results present several complex findings, e.g. differential consent rate, the influence of a single provider pair (explored in the sensitivity analysis) and the predictors of attrition, that have warranted further reflection in the discussion:

DISCUSSION / CONCLUSION

The Discussion and Conclusion are strong. They directly address the complex challenges raised in the Results. The conclusions are appropriately balanced, avoid overstatement, and are framed with the methodological complexities in mind. However, this has left little room for highlighting the study’s importance to policy and practice. As stated in the feedback on the impact statement, the Discussion could explicitly connect the findings to their significant implications for MHPSS policy and practice, particularly how this evidence can inform training guidelines for lay providers and the design of scalable, tiered-care models in resource-limited settings.

TIDieR checklist: For your reference, I used the TIDieR checklist to assess reporting on the intervention and implementation. Please see below for itemised comments

1. BRIEF NAME: The interventions are clearly and concisely named “CPSS-B” (Common Elements Treatment Approach - Brief) and “CPSS-E” (Common Elements Treatment Approach - Enhanced).

2. WHY: The manuscript provides a good rationale for the core components of the CPSS-B intervention (Screening, Psychoeducation, Safety). It also offers a clear justification for the additional “cognitive coping” element in CPSS-E, grounding it in Cognitive Behavioral Therapy (CBT) principles and citing supporting research.

3. WHAT: Materials: The authors mention the use of several materials, including a “brief, validated mental health assessment,” “slides for visual engagement,” and a “CPSS facilitator guide.” While the materials are named, the manuscript does not specify how to access them (e.g. URL or appendix), which can aid replicability.

4. WHAT: Procedures: The procedures for both intervention arms are described in detail, including content, delivery process, and follow-up.

5. WHO PROVIDED: The study provides a comprehensive description of the providers, including their professional background, connection to the veteran community, and extensive training process (2-day practice-based training, 1-day research methods training, practice groups, and ongoing weekly supervision).

6. HOW: The mode of delivery is clearly stated: single-session workshops delivered online via Zoom to groups of 5-10 participants. Additional detail could be given on how group dynamics were managed or how fidelity was monitored.

7. WHERE: The intervention occurred online. The authors note relevant features, such as the need for participants to be in a private space and the availability of technical support, which is appropriate for an online intervention.

8. WHEN and HOW MUCH: The study period is defined (September 2020 - February 2022). It is stated that both interventions were delivered in a single session. The CPSS-E session was approximately 30 minutes longer than the CPSS-B session. The manuscript does not specify the total duration of the CPSS-B workshop,

9. TAILORING: The follow-up call for the CPSS-E group included a review of their practice and additional instruction, which represents a planned, personalised element of the intervention.

10. MODIFICATIONS: The authors clearly describe two significant modifications: the switch from in-person to online delivery due to the COVID-19 pandemic (including the rationale and resulting adjustments) and the extension of the follow-up window for the treatment engagement outcome based on pragmatic observations.

11. HOW WELL (Planned): The plan to maintain and assess fidelity is described. Strategies included practice-based training, the use of a facilitator guide, and weekly supervision by experienced trainers. The “Sensitivity analyses” section explicitly mentions that “fidelity concerns identified in supervision” prompted further analysis, confirming that supervisors assessed fidelity.

12. HOW WELL (Actual): The sensitivity analysis details how fidelity concerns with one provider pair were identified and addressed. However, information on adherence across the rest of the provider teams is not as clear.

---

## [Reviewer Report]

No comments - all issues addressed as far as I can tell. I look forward to seeing the paper published. Thanks.

---

## [Reviewer Report]

Thank you for the opportunity to review again. This version is much more clear, and with additional clarity provided on the intervention itself. This article will significantly strengthen the evidence on MHPSS support for individuals in LMICs and humanitarian contexts.

One small revision - rather than “native Ukrainians” - first appearing on p 7, line 3 and elsewhere - I suggest “national Ukrainians” or just “Ukrainians”